# Capturing Conversational Gestures for Embodied Conversational Agents Using an Optimized Kaneda–Lucas–Tomasi Tracker and Denavit–Hartenberg-Based Kinematic Model

**DOI:** 10.3390/s22218318

**Published:** 2022-10-29

**Authors:** Grega Močnik, Zdravko Kačič, Riko Šafarič, Izidor Mlakar

**Affiliations:** Faculty of Electrical Engineering and Computer Science, University of Maribor, Koroška c. 46, 2000 Maribor, Slovenia

**Keywords:** conversational gestures, 3D gestures, motor skills, gesture reconstruction, kinematics, embodied conversational agents, Kanade–Lucas–Tomasi tracker, Denavit–Hartenberg

## Abstract

In order to recreate viable and human-like conversational responses, the artificial entity, i.e., an embodied conversational agent, must express correlated speech (verbal) and gestures (non-verbal) responses in spoken social interaction. Most of the existing frameworks focus on intent planning and behavior planning. The realization, however, is left to a limited set of static 3D representations of conversational expressions. In addition to functional and semantic synchrony between verbal and non-verbal signals, the final believability of the displayed expression is sculpted by the physical realization of non-verbal expressions. A major challenge of most conversational systems capable of reproducing gestures is the diversity in expressiveness. In this paper, we propose a method for capturing gestures automatically from videos and transforming them into 3D representations stored as part of the conversational agent’s repository of motor skills. The main advantage of the proposed method is ensuring the naturalness of the embodied conversational agent’s gestures, which results in a higher quality of human-computer interaction. The method is based on a Kanade–Lucas–Tomasi tracker, a Savitzky–Golay filter, a Denavit–Hartenberg-based kinematic model and the EVA framework. Furthermore, we designed an objective method based on cosine similarity instead of a subjective evaluation of synthesized movement. The proposed method resulted in a 96% similarity.

## 1. Introduction

Visual articulation of information through embodied behavior plays an important role in spoken social interaction [1]. Speech and gestures (including hand gestures, facial expressions, posture, and gazing) originate from the same representation, but are not necessarily based solely on the speech production process; i.e., “speech affects what people produce in a gesture, and that gesture, in turn, affects what people produce in speech” [2] (p. 260). In fact, more than 50 percent of visual articulation (i.e., embodied behavior) in spoken interaction adds non-redundant information to the common ground of the conversation [3]. Moreover, over 70% of the social meaning of a conversation or an interaction is transmitted through concepts other than words [4] (p. 27).

Recently, face-to-face interaction has been gaining attention, especially in interfaces where personalization is one of the key drivers, such as eHealth [5] and support for the elderly in their interaction with information technology [6]. Face-to-face interaction has been shown to elicit user engagement and stimulate the use of conversational interfaces [7], where the non-verbal, visual components drive the elicitation of affect and social awareness in human partners [8]. Overall, embodied conversational agents (ECAs) are becoming indispensable tools in personalizing and personifying everyday scenarios, where non-verbal behavior plays a crucial role in both representations of information and its understanding [9].

However, animating ECAs, whose non-verbal behavior is perceived as believable, is quite challenging, especially considering the complexity of the underlying bio-mechanical system [10]. The perceived believability of synthetic behavior and its plausibility depend on appearance, awareness, personality, emotional state, liveliness, illusion of life, consistency, diversity, and social fluency [11]. The dis-synchrony (unnaturalness) between verbal and non-verbal elements is most noticeable in synthesized non-verbal forms (i.e., shapes and poses), and especially through kinetics and “prosody” (i.e., fluidity, internal dynamics, movement phases, etc.) synthesis of movement [12].

Thus, two main challenges exist for the synthesized multimodal gestures to be perceived as believable [13]. The first challenge, i.e., the ‘symbolics’ of gesture, is related to the contextual alignment of visualized ‘shapes’ to speech and situational context, i.e., determining what kind of shapes a character should display in the given ‘semantic’ context: ‘What to display to visualize the given communicative intent?’. The symbolic alignment is, in general, implemented as a rule-based [14] or as a conversational behavior generation system with a machine learning (ML) baseline [15,16]. The second challenge, i.e., the ‘prosody’ gesture, is then related to the inner and outer fluidity (dynamics) of physical realization (animation) of selected visualizations, i.e., determining the trajectories and fluidity of movement in the given “non-semantic” context: “How to display the sequence of shapes to match conversational intent, i.e., acoustic and linguistic properties of spoken content?”.

The first challenge we tackled successfully in [16]. To tackle the second challenge, a wide range of techniques (data- and prosody-driven approaches) were introduced, to cope with significant requirements related to the believable fluidity of non-verbal expressions [17]. The main drawback of data/speech/prosody-driven approaches is that they are generated based on a small set of signals related mainly to the speech signal (e.g., pitch and prosody). Thus, they cannot facilitate “symbolics” [18].

The main idea behind the proposed method is to create contextually relevant resources that can be re-used when an embodied conversational agent generates a viable conversational sequence. The verbal and non-verbal context of an observed sequence, to be fed to behavior planning, is pre-annotated, and the role of the proposed method is to extract a possible visual articulation, including inner fluidity. Overall, systems utilizing gesture templates (e.g., procedural/physical animation) show the capacity to align movement with momentary context, as well as the context in the planned near future [19]. However, synthetic gestures still lack believability. To cope with the challenge of addressing the liveliness, diversity, and consistency of synthetic gestures adequately, we propose to exploit gesture tracking and 3D reconstruction to deliver a system capable of recording gestures expressed in the video during face-to-face conversation automatically, and storing them as gesture prototypes of the so-called “motor skills” [20]. We built the concept based on the following assumption: ECAs with diverse sets of resources, which preserve the dynamics and complexity of human movement, will be more successful in their attempt to mimic human-like conversational behavior. Such entities will be perceived as more believable virtual entities with human-like (and not human) attributes. Instead of subjective evaluation through human observation, we implement an objective measure to evaluate the naturalness of synthesized movement based on cosine similarity.

With the goal of reconstructing conversational gestures as natural as possible, we present our choice of suitable methods, announced in the title of the paper and our successful connection of stated methods in an efficient conversational gestures reconstruction system. We propose a measure based on cosine similarity for objectively evaluating the naturalness of synthesized hand movements generated by the proposed method instead of subjective evaluation through human observation, which is what, to the extent of our knowledge, was being done to evaluate gestures until now. In addition, we present the results of our system that were evaluated objectively on an embodied conversational agent called EVA (An EVA is an embodied conversational agent, developed in the Laboratory for Digital Signal Processing, Faculty of Electrical Engineering and Computer Science, University of Maribor [21]).

The paper is structured as follows. Section “Related Work” provides an overview of related work on gesture tracking and reconstruction techniques. Section 2 outlines the formalism used to implement the proposed tracking and reconstruction system. Section 3 shows the results of the system evaluated objectively on an embodied conversational agent EVA [21]. Section 4 provides the discussion, and the conclusions follow.

### Related Work

Synthetic responses are often perceived as lacking in consistency, diversity, and vividness. Gestures perceived as “random”, “non-aligned”, or “no-sense” distort the perception of eloquence, competence, human-likeness, and vividness of conversational responses. Creating believable movements is challenging, since the movements must be meaningful and natural, reflecting the coupling between gestures and speech [22]. In some cases, it has been suggested that having no gestures will achieve better persuasion and perceived believability than inadequate gesturing [23]. The main drawback most systems face in the context of believability is how to cope with diversity, i.e., how to create a repository of non-verbal expressions large enough to adequately represent both intent and thought [21].

Much of the complexity associated with the reconstruction process is related to the “posing” phase, in which the animators must handle a large number of on-screen handles associated with the character’s virtual skeleton. These handles allow direct or indirect modeling of the available degrees of freedom (DOFs) of all the individual character’s joints [24]. Moreover, to generate believable animation using CAD environments, the animators must understand and tackle many modeling and animation techniques (e.g., polygonal modeling, modeling with NURBS or subdivision, UV skinning, forward and inverse kinematics, rigging, etc.). The resulting animations are realistic in a given context. However, they may decrease believability significantly when the internal fluidity (dynamics) is changed due to different intent or prosodic context. To increase the versatility of the gestural morphology and to decrease the complexity of designing the movements a conversational agent can reproduce, we propose to build a 3D corpus of gesture prototypes [20]. For the modeling environment, we exploited the DAZ3D studio, which simplifies the CAD controls, and, through a vast repertoire of conversational resources, minimizes the human animator’s need for ‘artistic’ skills. Each prototype is expressively adjustable, and the EVA realizer [21] can mitigate the general issues of fluidity. However, since prototypes are generated manually by observing ‘real-life’ conversational expressions, the inner fluidity and the trajectory are not captured, but rather defined artificially through a set of orientational in-between points used to model the forward kinematics. As a result, the increasing complexity of gestures decreases the observed fluidity and perceived believability significantly.

To capture and preserve a high resolution of inner fluidity, performance-driven animation can be exploited to create 3D resources [25]. In this concept, an actor’s physical performance is transferred interactively to the virtual character to be animated. The method requires specialized equipment (e.g., a sensor suit) and specially trained experts, making it highly expensive and less suitable for non-professional animators. The mapping between the performer’s and character’s motion is also a complex task, since both entities operate in different spaces. Thus, the process requires sophisticated configuration steps and automatic retargeting [26]. In [27], the authors outline a sophisticated system consisting of multiple cameras and passive sensors to compensate for the lack of naturalness and capture movement generated during the conversation. Most multi-view methods utilize multiple cameras and exploit the image depth and shape from silhouette cues to capture the moving actor [28], or reconstruct gestures via a multi-view photometric stereo approach [29]. These methods typically require a high-resolution scan of the person as an input.

With advances in deep learning (DL) and image processing and the availability of depth camera sensors, new opportunities arise that could enable end-to-end reconstruction and capture of 3D resources. Methods integrating Kinect or similar depth sensors [30,31,32] or multi-view data [33,34] achieve impressive reconstructions, but do not register all frames to the same canonical template, and require complicated capture setups. Moreover, to represent conversational movement viably, the captured resources must originate from real-life situations integrating spontaneous behavior [35]. Recreation, even when performed by professional actors, will always reflect artificialness, resulting in less spontaneous and less diverse responses [36].

If we want to create a sufficiently large inventory of gestures that will enable the generation of natural gestures in interaction, and if we want to achieve a time-efficient generation of such an inventory, it makes sense to use a multitude of existing video recordings, and, consequently, it makes sense to use a method based on the use of video materials recorded with one camera. Most conversational corpora consist of TV interviews and theatrical plays that have shown themselves to be an appropriate resource of spontaneous conversational expressions, and are significantly more suitable for research in wider ‘discourse concepts’ than any artificially recorded material [37]. Most methods related to 3D Pose and Shape Estimation from monocular sources refer to (Deep) Convolutional Neural Networks ((D) CNNs) and leverage 2D joint tracking and predict 3D joint poses in the form of stick figures [38,39,40,41,42]. The major challenge with deep learning and similar probabilistic approaches is that the tracking process involves predicting the most probable configuration of the artificial skeleton. Thus, the captured conversational movement will approximate something known to the model rather than an exact replication of what is observed. Overall, the DNN-based approaches work well within the constraints of the known context (i.e., a fixed environment and known classes). However, in uncertainty, the models tend to underperform and require retraining [43]. The inconsistency and uncertainty of deep models (e.g., Pose Net, Open Pose) in many cases result in issues such as incoherence in fluidity (e.g., sudden shifts) and over smoothing of actual movement [44], leading to a decrease in believability when replicated as part of synthetic conversational behavior.

With our main motivation in mind, i.e., to capture conversational expressions from monocular video as similar to the original as possible, and by preserving the ‘prosody of movement’, i.e., fluidity and dynamics, we designed a novel system, which consists of a Kanade–Lucas–Tomasi tracker (KLT) [45] to track the observed body parts based on optical flow, and a Denavit–Hartenberg-based kinematic model [46] to reconstruct tracked features as 3D templates and store them as part of the EVA’s [20] motor skills repository. However, as with any image processing algorithm, mismatches in either tracking or reconstruction will always appear. Thus, in addition to the non-predictive method, it is crucial to have an objective measure to evaluate this effect. Instead of subjective evaluation of believability through perceptive experiments, as generally utilized in the field of embodied conversational agents, we propose a new method that allows for easy assessment of the mismatch generated in the tracking and reconstruction process.

## 2. Materials and Methods

### 2.1. Materials

The material we used in our work is a video signal with FullHD resolution and a different density of frames per second (frame rate). To test our system, conversational gestures were recorded with a video camera in a laboratory environment with a relatively impoverished background with only one actor. The resolution of the laboratory videos was FullHD (1920 × 1080) with H.264 compression and a frame rate of 30 FPS. In addition to laboratory videos, video clips from a video podcast were also selected; their content included spontaneously created conversational gestures and a diverse conversation with two performing actors. We used video content with a large number of spontaneously generated gestures. Such video content usually consists of videos with conversations. It is necessary to be aware that certain video content with professional actors (talk shows, evening news, etc.) does not offer a large amount of spontaneous and/or naturally created gestures. Professional actors know how to create conversational gestures that are not created spontaneously, but are acted out. We subjectively selected video content with gestures created spontaneously as our experimental example. The podcast videos were streamed from a social network, where they were published with the purpose of sharing video content. The resolution of the obtained video was also FullHD (1920 × 1080) with a different frame rate (25 FPS) than the laboratory video sources. Between 10 and 19 conversational gestures for each type of movement were analyzed, to evaluate the conversational gesture reconstruction from the EVAPose system. The analysis was performed for laboratory and spontaneously generated conversational gestures. We captured the spontaneous gestures from the video podcast, Gospoda [47].

### 2.2. Laboratory Set-Up

The core idea of the proposed method is to capture conversational expressions from different human collocutors engaged in interaction contained in the EVA-Corpus Video dataset, and store them back as “motor-skills” in the EVA-Corpus MotorSkills dataset, i.e., 3D artefacts to be re-used by embodied conversational agents during human-machine interaction. The workflow with individual steps is outlined in Figure 1.

The input to the proposed method was a color video stream, a conversational sequence contained in the EVA Corpus Video dataset. In the preparation phase, the best-fitting tracking points are selected automatically. The tracked points were filtered to reduce noise and possible inaccuracy in tracking—visualized as ‘jitter’ or sudden and instant jumps of the observed object from one position to another. The tracked geometry is sent to the Denavit–Hartenberg-based kinematic model and transformed into (Reconstruction in Figure 1) Euler angles (yaw, pitch, and roll) stored as a procedural animation (Create Resource in Figure 1). To validate the captured conversational expression (including the articulated shapes and inner fluidity) and compare it against the original, the expression was synthesized on our proprietary ECA realizer [21] by its in-scene recorder (Animation and Capture in Figure 1) functionality. If the synthetic system is recognized as similar (similarity index above 70%), the realization is registered as a possible visualization of the conversational concept. The following sections highlight the individual steps in more detail.

### 2.3. Preparation Step

Tracking arbitrary objects consistently and accurately in video sequences is challenging. Selecting robust features that best correspond to physical points and can, at the same time, be tracked well (e.g., mitigate occlusions, disocclusions and features that do not correspond to points in the world) is the first step in delivering an effective tracker. Shi-Tomasi’s implementation [45,49] represents a robust method to select “good features” and can, at the same time, compensate for lack of naturalness if and when these features are lost due to occlusion or “loss of visibility”; a common occurrence when tracking the movement of hands in multiparty discourse. Unlike the Harris Corner Detector, the Shi-Tomasi implementation proposes a variation in the selection of corners, and proposes a pixel to be considered as a corner by comparing the eigenvalues, i.e.:(1)$$R=\min\left(\lambda_{1}{,\lambda }_{2}\right)>\;\lambda ,$$
where $\lambda_{1}$ and $\lambda_{2}$ are two eigenvalues of a symmetric matrix and λ is the predefined threshold. The pixel is considered a corner when both $\lambda_{1}$ and $\lambda_{2}$ are above the threshold. Figure 2 highlights the selection of the N strongest corners as defined by Shi-Tomasi (a) and selected tracking points (b), to be tracked and used as input in the reconstruction.

As outlined in Figure 2, the algorithm operates over grayscale images. The “quality” of corners (i.e., the λ threshold) is specified as a value between 0 and 1. All the corners below the threshold are rejected. Since we wanted to track only specific artefacts representing the shoulder, elbow, and wrist “joints”, the user selects the regions of interest. Based on the manual selection of the region of interest, the algorithm selects the strongest corner automatically (i.e., the “green” circles in Figure 2b) as the final tracking point, and “rejects” all nearby corners of interest. In the tracking process, the tracking points are regarded as features.

### 2.4. KLT Feature Tracker

The KLT feature tracker [50] computes the displacement of features between consecutive frames by aligning a second image *J* to an input image *I*, where *I*(*x,y*) represent the intensity of the image at [*x y*]^T^.

Let:(2)$$u={\left[u_{x}\;u_{y}\right]}^{T},$$
where u represents the point at coordinates $\left(x,y\right)$ in the first image I. The goal of tracking is to find point v in the second image  J, where the displacement d is minimal; thus $I\left(u\right)\;$and $J\left(v\right)$ are similar:(3)$$v=u+d={\left[u_{x}+d_{x}\;{\;u}_{y}+d_{y}\right]}^{T}.$$

The displacement $d={\left[d_{x}\;d_{y}\;\right]}^{T}$ represents the image velocity (optical flow) at u. The minimal difference is computed as the mean squared error function:(4)$$\varepsilon \left(d\right)=\overset{u_{x}+w_{x}}{\underset{x=u_{x}-w_{x}}{\sum }}\overset{u_{y}+w_{y}}{\underset{y=u_{y}-w_{y}}{\sum }}{\left(I\left(x,y\right)-J\left(x+d_{x},\;y+d_{y}\right)\right)}^{2}$$
where $w_{x},w_{y}$ represent the integration window size parameter of the template window of size $\left(2w_{x}+1\right)\times \left(2w_{y}+1\right)$.

The intensity of the image is represented by a small template window of n×n, centered at one of the feature points. $J\left(x\right)$ is the same window in the next frame. d represents the displacement vector, and η represents the error introduced due to the shape change.

During the tracking process, the goal is to find point v in image J  that corresponds to point u in the image I:(5)$${\bar{v}}_{opt}=G^{-1}\bar{b},$$
where:(6)$$G\;\dot{=}\;\overset{p_{x}+w_{x}}{\underset{x=p_{x}-w_{x}}{\sum }}\overset{p_{y}+w_{y}}{\underset{y=p_{y}-w_{y}}{\sum }}\left[\begin{array}{cc} I_{x}^{2} & I_{x}I_{y}\\ I_{x}I_{y} & I_{y}^{2} \end{array}\right]\;$$

*I_x_* and *I_y_* represent image derivatives.

The image mismatch vector $\bar{b}$ is defined as:(7)$$\bar{b}\dot{=}\;\overset{p_{x}+w_{x}}{\underset{x=p_{x}-w_{x}}{\sum }}\overset{p_{y}+w_{y}}{\underset{y=p_{y}-w_{y}}{\sum }}\left[\begin{array}{c} \delta I\;I_{x}\\ \delta I\;I_{y} \end{array}\right]$$
δI represents the image difference. In standard optical flow computation, the goal is to find ${\bar{v}}_{opt}$ as displacement $\bar{v}$, which minimizes the matching function $\varepsilon \left(\bar{v\;}\right)$:(8)$$\varepsilon \left(\bar{v}\right)=\varepsilon \left(v_{x},\;v_{y}\right)\overset{p_{x}+w_{x}}{\underset{x=p_{x}-w_{x}}{\sum }}\overset{p_{y}+w_{y}}{\underset{y=p_{y}-w_{y}}{\sum }}{\left(A\left(x,y\right)-B\left(x+v_{x},\;y+v_{y}\right)\right)}^{2}$$
where optimum $\bar{v}\;$ is calculated through Taylor expansion:(9)$${\left.\frac{\partial \varepsilon \left(\bar{v}\right)}{\partial \bar{v}}\right|}_{\bar{v}=\bar{v_{opt}}}=\frac{1}{2}\left[\frac{\partial \varepsilon \left(\bar{v}\right)}{\partial \bar{v}}\right]\;\approx G\bar{v}-\bar{b}$$

Because of the first-order Taylor approximation, this is only valid when the pixel displacement is small. Thus, the standard optical flow computation is performed in k steps and defined by:(10)$$\bar{v}=d^{L}={\bar{v}}^{K}=\overset{K}{\underset{k=1}{\sum }}{\bar{\eta }}^{k}$$
where $\bar{v}$ represents the final optical flow and $d^{L}$ the displacement, K the number of iterations to reach convergence, and new pixel displacement (i.e., one step in the LK optical flow computation) ${\bar{\eta }}^{k}$ is defined as:(11)$${\bar{\eta }}^{k}=G^{-1}{\bar{b}}_{k}$$

Derivatives Ix, Iy in the image mismatch vector are computed at the beginning, and only δI is recomputed at each step k. The overall iteration completes when ${\bar{\eta }}^{k}$ is smaller than the threshold, or the maximum number of iterations is reached.

The implementation of the KLT tracker used in our research is highlighted in Figure 3. The preparation step, definition of feature points and tracking points according to the defined process, is described in Section 2.1. Feature points were selected according to Shi-Tomasi’s approach, and the tracking points were set to regions representing the “shoulder”, “elbow”, and “wrist” joints. We used a 5×5 points integration window.

The tracking process implements the iterative optical flow computation and matches tracking features between the current $image_{i}\;$*(*J*)* and the previous $image_{i-1}$
*(I)* by tracking feature points, where i is on the interval [1, *n*] and n is the last frame of the video recording of the conversational sequence. If point v in *image_i_* that corresponds to point u in $image_{i-1}$ cannot be found (i.e., the feature falls outside of the image), and/or the image path around the track point varies too much (i.e., the cost function is larger than a threshold), the feature point is regarded as lost and is deleted. To recreate the expression, all tracking points must be registered in all frames, and the missing feature points must be replaced. The process exploits the Shi-Tomasi detector to create a new set of good features to replace lost features. The tracking points are relocated automatically to the closest new best feature point. The new feature points and 2D coordinates of each tracking point are saved, and the algorithm may proceed with the next frame. The tracking process completes when the last frame of the video is reached.

### 2.5. Filtering

The designed tracker implements “tracking-by-detection” and does not implement feature descriptors, such as SIFT [51]. This means that the tracking accuracy varies depending on the rotation, scale, and image perspective distortions (including lighting changes). The inaccuracy results in a “long-distance” move of a tracking point instead of a small shift in position (i.e., noise). While reconstructing the movement on the ECA, the jumps will be observed as instant “jumps”—movements which cannot be expected in real life. To avoid this, we implemented a digital filter based on the Savitzky–Golay algorithm [52]. The Savitzky–Golay filter belongs to the family of FIR filters, and provides an estimate of the derivative of the smoothed signal using convolutional sets derived from least-squares formulas coefficients. Savitzky–Golay filters minimize the least-squares error in fitting a polynomial to a sequence of noisy data. Consequently, the precision of data increases without distorting the signal tendency. Thus, the method is suitable for signal smoothing [53,54].

Let us consider the captured tracked points as a compositum of captured movements, i.e., the main signal $f\left(l\right)$, corrupted randomly by distortions, i.e., $w\left(l\right)$, thus the real signal (stream of tracking points) is defined as:(12)$$x\left(l\right)=f\left(l\right)+w\left(l\right),\;\;l=0,\ldots ,\;L$$
where $x\left(l\right)$ indicates the lth tracking point (i.e., the lth frame) in the signal with L points (sequence of data). The goal is to smooth the $x\left(l\right)$ to reduce the level of the remaining noise to as low as possible, i.e., to minimize the following MSE:(13)$$\varepsilon_{n}=\overset{M}{\underset{i=-M}{\sum }}{\left(P\left(i\right)-x\left(i\right)\right)}^{2}=\;\;\overset{M}{\underset{i=-M}{\sum }}{\left(\overset{n}{\underset{k=0}{\sum }}a_{k}i^{k}-x\left(i\right)\right)}^{2}$$
where the smoothing is carried out with a symmetric window with width N=2M+1 samples around the “reconstruction point”. In this case, smoothing can be represented as a polynomial with the order $n\left(P\left(i\right)\right)=\overset{n}{\underset{k=0}{\sum }}a_{k}i^{k};k=0,\ldots ,\;n,\;$ and $a_{k}$ is the kth coefficient of the polynomial.

The filter output is then equal to the value of polynomial (n) in the central point $y\left(0\right)$; $y\left(0\right)=p\left(0\right)=a_{0}$. To calculate the next point, the window N is shifted by 1 unit. Savitzky and Golay [52] showed that the above process of ‘filtering’ is equivalent to convolving samples in windows with a fixed impulse response:(14)$$y\left(k\right)=\overset{M}{\underset{i=-M}{\sum }}w_{i}x\left(k-1\right)$$

To select the SG parameters used for smoothing optimally, we applied the power spectrum analysis. A power spectrum analysis was performed on an arbitrarily selected area of an unfiltered input signal. The analysis showed the level of the signal power spectrum and noise. Such analysis was also calculated for a filtered input signal with SG parameters of choice. In the area of the high frequency harmonics, we checked at what distance from the densification of the signal spectrum there was still a greater change in the spectrum of the individual filtered signal at the noise level. It was found that, with a window width of N = 9 and a polynomial degree 3, we achieved a sufficient smoothing effect. Figure 4 shows the power spectrum of the smoothed signal, the smoothed signal with a third-degree polynomial, and a window width of 9 is highlighted. It can be seen that the level of the power spectrum of the smoothed signal is less intense at slow transitions of the input signal than at faster ones. From this, we can conclude that the parameters selected preserve the signal’s slower jumps, while the smoothing increases in the areas with faster jumps (a part of the signal with a higher frequency). Figure 5 shows the smoothing results using an SG filter with a window width N = 9 and a polynomial of the 3rd order. For the objective weight function w, a cubic function was used.

### 2.6. Denavit–Hartenberg Based Reconstruction

As highlighted in Figure 1, the tracking points are sent to the reconstruction phase. In the first step, internal angles are calculated from tracking points using basic angle functions based on the player’s position in the video scene. The internal angle is calculated between the starting point $o_{a}$ (in our case the player’s shoulder on the video signal) and the end of the player’s arm (end-effector) $o_{ef}$. In the case when the player is facing us, the internal angle is calculated as $q_{8}=\arctan2\left(o_{a},\;o_{ef}\right)$. For each joint that rotates, we can write $q_{i}=\arctan2\left(o_{a},\;o_{i}\right)$. Signal tracking points for each joint are marked as $o_{i}$.

In this phase, 2D coordinates of tracked points are converted into Euler angles, which can be animated by our conversational agent. The arm is deconstructed into a manipulator consisting of three spherical joints: the shoulder, the elbow, and the wrist joint. Figure 6 outlines the designed manipulator. The end-effector is placed at the far end of the arm (e.g., the tip of the hand) as a reference point used in the automatic kinematic analysis algorithm.

The goal is to determine the rotation of each joint in the mechanism of human arm movement from the positions of the tracking points. As outlined in Figure 6, the proposed kinematic model assumes each spherical joint is represented by multiple revolute joints that permit linear motion along a single axis. Using a single degree of freedom allows us to represent each angle of rotation of the spherical joint with a single real number, and the rotation in the spherical joint as a composition of single-axis rotation. This allows us to determine the position, and, more importantly, the orientation of tracking points in a systematic way, where the cumulative effect ($A_{i}$) is calculated using the Denavit–Hartenberg (D–H) convention [46].

We assumed the proposed model consisted of eight revolute joints and ten links. We assumed that $joint_{i}$ connects $link_{i-1}$ with $link_{i}$. Thus, when $joint_{i}$ was actuated, $link_{i}$ and further links in the kinematic chain of the robot arm moved. To perform the kinematic analysis, a coordinate frame was attached to each link, i.e., $x_{i}y_{i}z_{i}$ to $link_{i}$, represented by the tracking point. Using the D–H convention, we assumed $A_{i}$ was a homogeneous transformation matrix which expresses the position and orientation of $x_{i}y_{i}z_{i}$ in respect to $x_{i-1}y_{i-1}z_{i-1}$. The $A_{i}$ varied as the configuration of the manipulator was changed; however, since we assumed the use of revolute joints, $A_{i}$ is a function of a single joint variable and can be represented as the product of four basic transformations, rotation, and translation around the $z_{i-1}$ and $x_{i}$ axes:(15)$$A_{i}^{i-1}=Trans\left(z_{i-1},d_{i}\right)Rot\left(z_{i-1},\;\theta_{i}\right)Trans\left(x_{i},a_{i}\right)Rot\left(x_{i},\;\alpha_{i}\right)=\;=\begin{array}{ccccc} \left[\begin{array}{cccc} 1 & 0 & 0 & 0\\ 0 & 1 & 0 & 0\\ 0 & 0 & 1 & d_{i}\\ 0 & 0 & 0 & 1 \end{array}\right] & \left[\begin{array}{cccc} c\theta_{i} & -s\theta_{i} & 0 & 0\\ s\theta_{i} & c\theta_{i} & 0 & 0\\ 0 & 0 & 1 & 0\\ 0 & 0 & 0 & 1 \end{array}\right] & \left[\begin{array}{cccc} 1 & 0 & 0 & a_{i}\\ 0 & 1 & 0 & 0\\ 0 & 0 & 1 & 0\\ 0 & 0 & 0 & 1 \end{array}\right] & \left[\begin{array}{cccc} 1 & 0 & 0 & 0\\ 0 & c\alpha_{i} & -s\alpha_{i} & 0\\ 0 & s\alpha_{i} & c\alpha_{i} & 0\\ 0 & 0 & 0 & 1 \end{array}\right] & =\left[\begin{array}{cccc} c\theta_{i} & -s\theta_{i}c\alpha_{i} & s\theta_{i}s\alpha_{i} & a_{i}c\theta_{i}\\ s\theta_{i} & c\theta_{i}c\alpha_{i} & -c\theta_{i}s\alpha_{i} & a_{i}s\theta_{i}\\ 0 & s\alpha_{i} & c\alpha_{i} & d_{i}\\ 0 & 0 & 0 & 1 \end{array}\right] \end{array}$$
where $\theta_{i}$, $d_{i}$, $a_{i}$, $\alpha_{i}$ are parameters associated with $link_{i}$ and $joint_{i}$ denoted as joint angle, link offset, link length, and link twist. The parameters mentioned above are shown in Figure 6.

Since revolute joints were used, the $A_{i}^{i-1}$ is a function of a single variable $\theta_{i}$ and the other three parameters are denoted as D–H parameters and are constant for a given link. The D–H parameters for each link of the proposed kinematic model are shown in Table 1.

Where link lengths $a_{1},\;a_{5},-l_{x},-l_{y}$ and link offsets$\;d_{4},\;l_{y}$, were calculated as the average values of measurements performed over multiple human arms:(16)$$\left(a_{n},d_{n},l_{n}\right)=\left(\frac{\sum_{1}^{n_{p}}a_{n}}{n_{p}},\frac{\sum_{1}^{n_{p}}d_{n}}{n_{p}},\frac{\sum_{1}^{n_{p}}l_{n}}{n_{p}}\right)$$
where $n_{p}$ represents the number of candidates participating in the measurements ($n_{p}=10$) and the variable $l_{n}$ represents the end-effector position. For our case, we calculated the D–N parameters as $a_{1}=30\;cm\;$, $a_{5}=30\;cm\;,\;d_{4}=30\;cm$, $l_{x}=1,5\;cm$, $l_{y}=2.0\;cm$ and $l_{z}=12\;cm$.

Using the D–H parameters in Table 1, we calculated the homogeneous transformation matrices $A_{i}^{i-1}$ for each joint, and created a reference transformation matrix for the forward kinematics of the proposed kinematic model:(17)$$A_{9}^{0}=A_{1}^{0}A_{2}^{1}A_{3}^{2}A_{4}^{3}A_{5}^{4}A_{6}^{5}A_{7}^{6}A_{8}^{7}A_{9}^{8},$$
where the last two matrices $A_{8}^{7}$ and $A_{9}^{8}$ are the matrices of the end effector.

However, it is not trivial to represent any arbitrary homogeneous transformation using only four parameters [46,55]. Given two consecutive frames, 0 and 1, with given coordinate frames $x_{0}y_{0}z_{0}$ and $x_{1}y_{1}z_{1}$ respectively, we assumed there exists an $A_{0}^{1}$ homogeneous transformation matrix. Moreover, we assumed the axis $x_{1}$ was perpendicular to $z_{0},$ and $x_{1}$ intersects the $z_{0}$ axis. Under these conditions, there exist unique numbers a,d,θ,α, and Equation (15) can also be written as:(18)$$A_{i}^{i-1}=\left[\begin{array}{cc} R & p\\ \begin{array}{ccc} 0 & 0 & 0 \end{array} & 1 \end{array}\right]$$
where R describes the rotation matrix of joints and p describes the displacement. We can write the rotation in a homogeneous transform matrix as:(19)$$R_{i}=R_{i,z}\left(\gamma \right)R_{i,y}\left(\beta \right)R_{i,x}\left(\alpha \right)=\begin{array}{c} yaw\\ \left[\begin{array}{ccc} cos\gamma_{i} & -sin\gamma_{i} & 0\\ sin\gamma_{i} & cos\gamma_{i} & 0\\ 0 & 0 & 1 \end{array}\right] \end{array}\begin{array}{c} pitch\\ \left[\begin{array}{ccc} cos\beta_{i} & 0 & sin\beta_{i}\\ 0 & 1 & 0\\ -sin\beta_{i} & 0 & cos\beta_{i} \end{array}\right] \end{array}\begin{array}{c} roll\\ \left[\begin{array}{ccc} 1 & 0 & 0\\ 0 & cos\alpha_{i} & -sin\alpha_{i}\\ 0 & sin\alpha_{i} & cos\alpha_{i} \end{array}\right] \end{array}$$
where the angles α, β, and γ represent the Euler rotation. The names of each rotation angles are roll, pitch, and yaw. Since the embodied conversation agent EVA [21] has a default position different from our kinematic model, the calculated data must be adjusted to be suitable for the EVA-Script Template using a 90-degree rotation of α angle (roll):(20)$$R_{i}\;\overset{adjustment}{\to }\;R_{i,EVA}=R_{i,z}\left(\gamma \right)R_{i,y}\left(\beta \right)R_{i,x}\left(\alpha +90\right)$$

The above adjustment represents a method to normalize scene results to any given space of any embodied conversational agent with an underlying three joint-based skeletal structure. The calculated data can now be used for 3D conversational resources, as defined by the final component, i.e., the Resource Generator.

### 2.7. Resource Generator

In this step, the captured and reconstructed stream of Euler angles is transformed into a procedural animation, an EVA-Script Template compatible with the repository of “motor skills”. In the EVA Framework, a conversational expression or gesture is defined as a function of conversational intent and its realization through visualization (i.e., movement model) [16], i.e.:(21)$$\hat{G}=T_{m}^{-1}\hat{H}$$
where $\hat{G}$ represents gesture $T_{m}^{-1}$ contextual interpretation based on conversational intent, and $\hat{H}$ is the movement model used to “visualize” the conversational intent. The movement model is then defined as a transition between the pose at the beginning ($P_{S}$) and the pose at the end ($P_{E}$), via a trajectory (J), carried out over time t.
(22)$$\hat{H}=J\left(P_{S},\;\right.{\left.P_{E}\right)}_{|t}$$

Since the realization engine utilizes procedural animation and forward kinematics, the trajectory J is specified as a sequence of in-between frames, i.e., $J=\left\{P_{S+1},\;\ldots \;,\;P_{E-1}\;\right\}$. The attribute t  is used to “optimize” the number of in-between-frames given the time constraint t*,* and the realizers targeted frame rate f*:*(23)$$N=round\left(\frac{n_{J}+1}{t\times f}\right)$$

Here, N represents the number of in-between frames in J to skip, $n_{J}$ represents the total number of frames captured by J. We added 1 frame to preserve the number of in-between frames, since the first transformation from configuration $P_{S-1}$ to $P_{S}$ (i.e., the *start pose*) is captured by J. The t×f represents the maximum number of frames the realizer can implement without any impact on the inner fluidity. Namely, adding too many in-between frames will “slow” down the gesture synthesis. The remaining series of configurations are, afterwards, transformed into an EVA template [22]. The overall process is outlined by the following pseudocode:
**Algorithm 1.** Proposed algorithm for creating resources.*t = calculate_from timestamps**f = from_config**N = round_up* (*size*(*frame*) + 1)/(*t* × *f*)**if***N* > 1 **then**
*H = array (t* × *f)***else***H = array* (*size*(*frame*) *+ 2*)H [0] = to_unit(*frame* [0])**for***i* = 1 to *size*(*frame*) − 2:**if***i mod N ==* 0 **then:**append(*toUnit(frame[i]), H)***else:****continue***append*(*toUnit(frame[*size(frame) − 1*]), H)****createGesture* (*H*, *t*)**

The algorithm always takes the first and the last frame as the start and end poses, respectively, and every ith in-between frame, such as i mod N equals zero. Other frames are dropped. The function toUnit maps the 3D definition of each frame into the EVAScript’s unit notation, i.e., *{key, list}* pair where *key* represents the articulated joint (e.g., collar, shoulder, elbow, forearm, or wrist) and *list* represents a sequence of 3D configurations for the joint (i.e., the sequence of yaw, pitch, roll configurations) that constitute the 3D transformation from pose *i* to pose *j.* The function *createGesture* then defines $\hat{G}$ as a procedural fragment written in EVAScript markup as shown in Figure 7.

## 3. Results

In this section, we will present the content related to evaluating the reconstructed motion. Aspects will be presented of the evaluation of the reconstructed movement, evaluation procedures, and the characteristics used in such procedures. Here, we should keep in mind that the main objective is creating natural movement from sources whose content is naturally generated conversation. The existing corpora of reconstructed conversational gestures suffer from a relatively small number of artificially generated gestures. The use of social networks and their video content sharing allows us to obtain resources with the content of naturally generated conversational gestures. Although we can create a large number of reconstructed conversational gestures in this way, we have to evaluate these gestures objectively and subjectively. In our evaluation, we focused on objective evaluation, which allows us to remove poorly reconstructed gestures before evaluating them subjectively.

As already mentioned, the assessment of the similarity between actual and reconstructed movement in most cases comes from the context of use. In some cases, only the rough course of the movement may be of interest, while, in others, the accuracy of the reconstruction of the movement of each joint is necessary. Usually, a 3D reconstructed movement is considered to be similar to the real one if the only difference is a global geometric transformation such as translation or rotation, and the speed of conversational gestures can also be taken into account [56]. In the case of conversational gestures, such an aspect is not satisfactory, since logical and numerical similarities are missing. Aspects taking into account logical and numerical similarities define the logical similarities of movement as several versions of the same action or sequence of actions. In most cases, the algorithms used to evaluate logical and numerical similarities are based on quantitative characteristics [57]. However, it is necessary to consider that a logically similar movement can be numerically different, and vice versa. In many contexts of use, partial similarities are important, and we reconstructed the movement of some parts of the body differently from others. In this aspect, it is necessary to be aware that considering the extracted characteristics in the similarity measure of “unimportant” parts of the body can impact the result negatively [58].

In our case, the similarity between the input signal, tracked using the Kanade–Lucas–Tomasi tracker, and the reconstructed signal, tracked with the same tracker, was calculated using local similarity assessment features. This approach aimed to address the partial, logical, and numerical aspects of the similarity between reconstructed and actual human movement. The cosine distance features were used to compare the two signals on the entire time axis (marked as sim1) to assess the similarity. At the same time, the similarity of both signals was also assessed at selected time points, representing the moment of the conversational gesture position (*sim2*) with the greatest similarity. An important feature that must be considered in our case is the number of displayed images per second (frame rate) measured in FPS. With this feature, we added a new dimension to the similarity assessment, which captured the aspect of the global transformation with time to a certain extent. By taking into account the time feature, we can evaluate the directly reconstructed gestures quantitatively. We performed an experimental evaluation for two methods: OpenPose and our proposed EVAPose. Due to the fact that the OpenPose method failed to provide results for the original frame rate of the recording, we defined the condition sim2, which enabled us to compare the results of the methods at a similar frame rate. In this, we implemented the proposed EVAPose method in a way that it generated the same frame rate as the OpenPose method. A similarity assessment was made between the input video signal and the reconstructed signal using the proposed EVAPose and OpenPose systems. The similarity score marked with sim1 means the average of the scores of individual gestures. The sim2 rating contains the maximum value from the average of the ratings of individual gestures. Both systems measured both ratings. Similarity scores for the cases used in the evaluation are given in Table 2.

The results of the reconstructed motion from video clips created in the laboratory are shown in the upper part of Table 2. The content of such videos was created to test the system, and does not show naturally generated conversational gestures. The lower part of Table 2 shows the results of conversational gestures, which are gesticulated in a spontaneous and natural way. Table 2 lists one of the tested sources with such content, where the content was, in most cases, a content-varied conversation between two actors with different and spontaneous gestures. The language used by the two actors was Slovenian. The results show how the proposed system compared with the OpenPose system [42]. Conversational gestures containing the most movement in the vertical and horizontal directions were selected for comparison. We considered such types of conversational gestures as the geometric basis for more complex gestures, which consist of a combination of horizontal and vertical movements.

To assess naturalness, we monitored events that occurred during the reconstructed gesture. We noticed that the signal from the OpenPose system did not have a smooth continuous transition compared to the EVAPose system. At some moments, an unnatural movement occurred, triggered by a sudden jump in the angle of a single joint. To detect such jumps, we analyzed the signal by differentiating the reconstructed signal (position as a function of time) three times (Figure 8). The third derivative showed us the jerks (jumps) and their number. Figure 8 shows the third derivative of the reconstructed signal (position as a function of the number of samples) as a function of the number of samples.

## 4. Discussion

It can be seen from Table 2 that, in absolute terms, the similarity score using the OpenPose system was better than the score using the proposed system. The highest frame rate achieved by the OpenPose system was 11 FPS. To achieve comparability, we adjusted the frame rate of the proposed system by increasing the degree of the polynomial in the smoothing process (described in previous chapters). When we reached a frame rate of 19 FPS, the similarity did not improve anymore, but the computational complexity of the proposed method increased tremendously. The same gesture was reconstructed in all videos containing multiple vertical and horizontal movements. By reducing the frame rate in the EVAPose system, it was shown that the similarity had improved. In the case of horizontal movement from the video content created in a laboratory environment, the score exceeded that of the OpenPose system. The laboratory video content with a length of 46 s, which was given to the input of the proposed system, contained 10 gestures that were reconstructed completely successfully.

The results show that, despite the lower average score, a gesture was reconstructed better (in the sense of naturalness) with the proposed system than with the OpenPose system. The better reconstruction in terms of naturalness was manifested mainly in the jumps of the reconstructed signal. It can be seen from Figure 8 that the EVAPose system had a smaller number of jerks with a larger amplitude. It is also important that the EVAPose system jerks were not aligned with the OpenPose system jerks, which can undoubtedly be attributed to the fact that the jerks are not the result of naturally generated jumps in conversational gestures, but are a statistical error in the algorithms.

The results show that, for the real case from the Gospoda dataset, the proposed system did not reach the similarity level achieved by the OpenPose system; however, the results were comparable. On the other hand, the EVAPose system can maintain the frame rates. In the case with an adjusted frame rate, the video content with a lower frame rate was chosen only to test the EVAPose system at different frame rates. Here, it is worth emphasizing that the preliminary laboratory experiments in the subjective similarity evaluation showed that the objective assessment could reach the subjective level of assessment in all cases.

Despite the better evaluation, the reconstruction with the OpenPose system did not provide a sufficiently large degree of naturalness. Even though the gesture was captured from video content in which a spontaneously created conversational gesture appeared, the OpenPose reconstruction system introduced an error in the reconstruction of the rotation of individual joints, which a human cannot create. We concluded that the cause of such errors comes from the learning process in the learning technologies. Unlike our system, the OpenPose system does not have clearly defined areas of rotation of individual joints in the human body. In our case, we defined the rotation areas of individual joints using the Denavit–Hartenberg notation before calculating forward kinematics, thus ensuring that unnatural joint rotations did not occur. The choice of forward kinematics and the notation of the kinematic model according to Denavit–Hartenberg also allowed a continuous transition of a specific movement.

As already stated, the assessment of similarity is relative in nature, because the systems can be used in different ways, in some of which the similarity assessment is important, and computational complexity of gesture reconstruction does not matter, nor how long the reconstruction process takes. In other cases, however, we wanted to preserve the frame rate, and can settle for a lower similarity score of the speech gesture reconstruction. In this case, we must be aware that the frame rate was preserved at the expense of lower similarity, which was conditioned by the acceptability of the gestures generated in this way, and by the perception of their naturalness and other important characteristics of successful spoken social interaction.

## Figures and Tables

**Figure 1 sensors-22-08318-f001:**
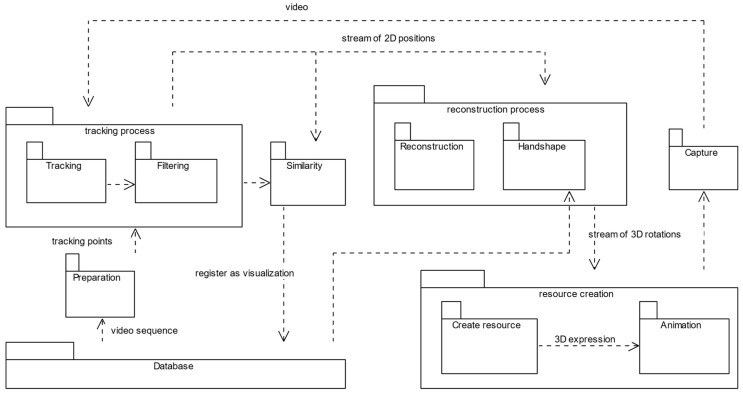
Workflow to capture conversational behavior in spontaneous discourse automatically, store it as gesture templates, and interconnect the captured templates with other verbal and nonverbal features of the observed sequence. The hand shape is not tracked; a CNN model was used to select the shape from a dictionary of possible shapes based on the HamNoSys notation system [48].

**Figure 2 sensors-22-08318-f002:**
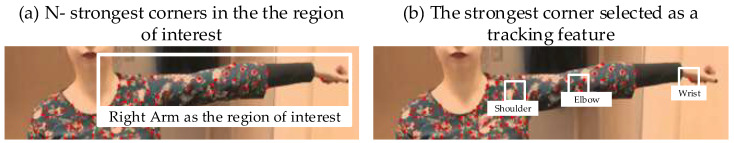
Example of (**a**) Good features as extracted by the Shi-Tomasi detector and (**b**) Tracking points as the strongest corners in a specific region, representing the “tracked” joints in the human skeleton.

**Figure 3 sensors-22-08318-f003:**
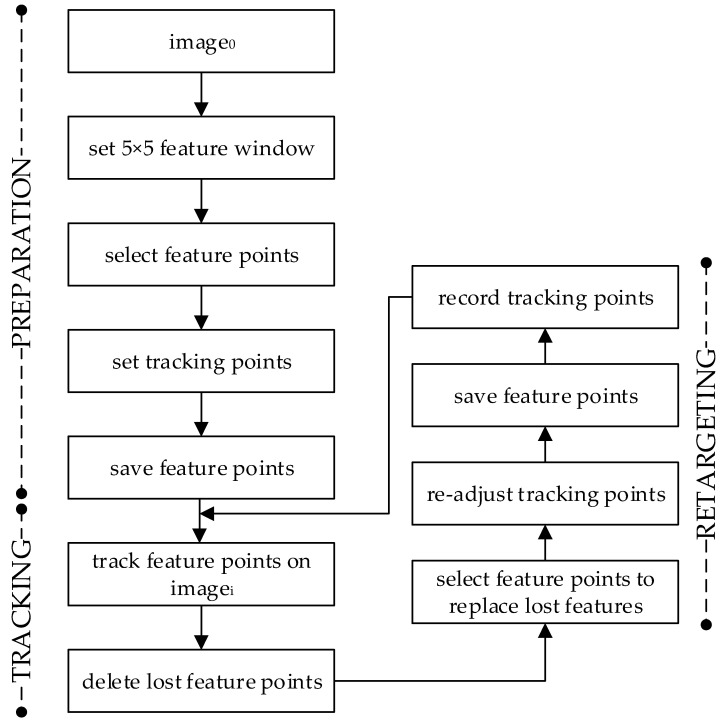
Overview of the implementation of the pyramidal KLT Tracker.

**Figure 4 sensors-22-08318-f004:**
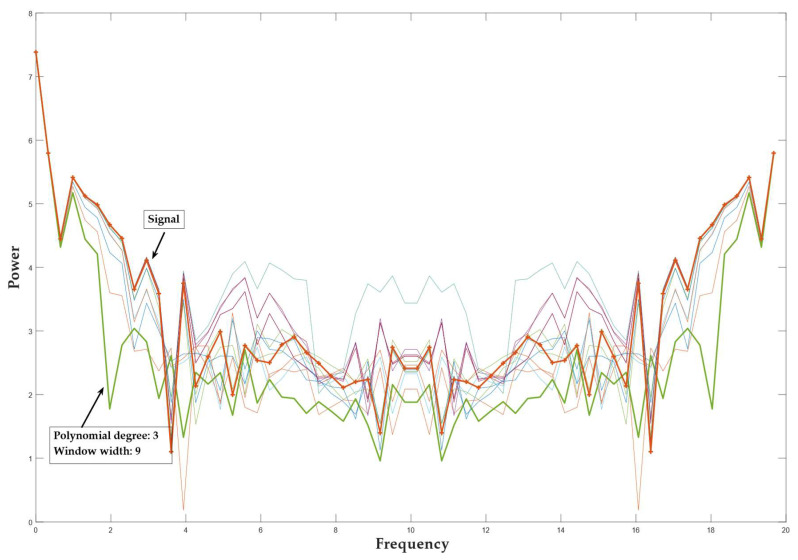
Power spectrum analysis of a filtered and nonfiltered signal. The green curve represents the optimal filtered signal; the orange curve represents the nonfiltered signal.

**Figure 5 sensors-22-08318-f005:**
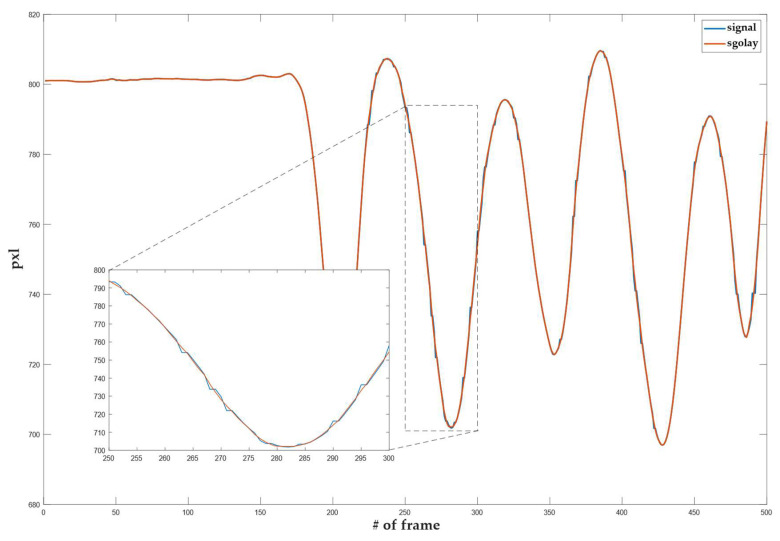
Smoothing the raw tracking results with an SG filter.

**Figure 6 sensors-22-08318-f006:**
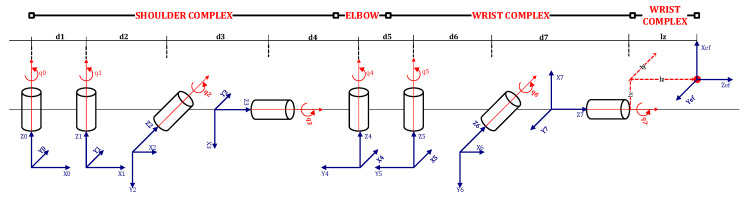
Visualization of the kinematic model, i.e., the arm manipulator consisting of complex cylindrical joints, implementing degrees of freedom of a spherical joint utilized in the skeleton of the realization entity.

**Figure 7 sensors-22-08318-f007:**
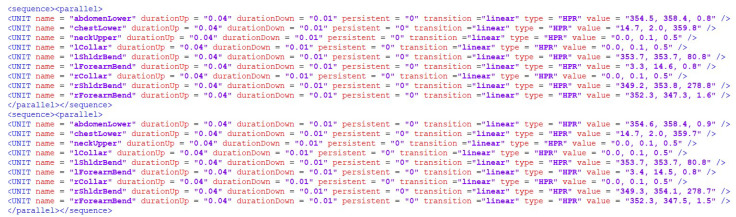
An example of procedural animation formulated in EVAScriptMarkup, each *<sequence><parallel>* represents configurations P_i_, P_i+1_ as the transition between two consecutive frames adjusted to the frame rate scaling. *durationUp* represents the duration of the transition, and is calculated as $\frac{sizeof\left(H\right)}{t}$ and the value represents the 3D configuration of the “joint” (movement controller) in Euler angles expressed in roll–pitch–yaw (HPR) notation.

**Figure 8 sensors-22-08318-f008:**
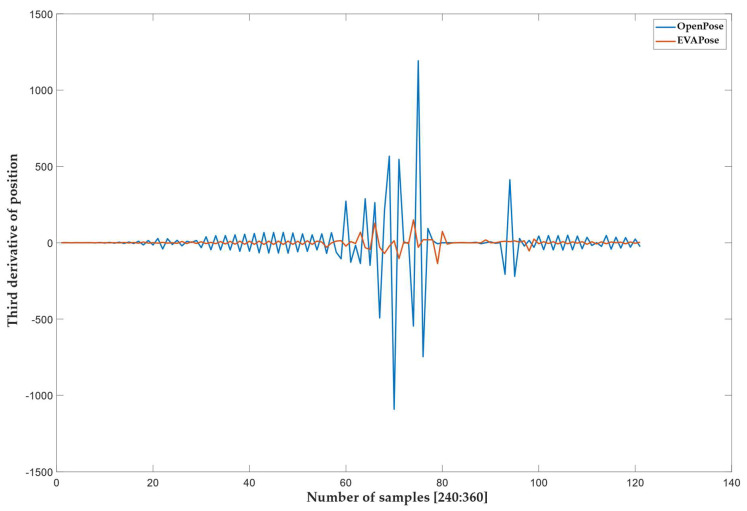
The red and blue curves represent the third derivative (jerk) of the reconstructed signal (position) from EVAPose and OpenPose, respectively. The third derivative is shown as a function of the number of samples. The number and amplitude of jerks in this type of analysis show unnatural and high−energy concentrated spikes on the reconstructed signal. Only a section of the entire signal (120 samples/1380 samples) is shown for easier and better presentation of the reconstructed signal’s third derivative.

**Table 1 sensors-22-08318-t001:** The Denavit–Hartenberg parameters for the kinematic model.

$Link_{i}$	$a_{i}$	$d_{i}$	$\alpha_{i}$	θ
1	$a_{1}$	$a_{1}$	π2	$q_{1}$
2	0	0	−π2	$q_{2}$
3	0	0	π2	$q_{3}$
4	0	0	−π2	$q_{4}$
5	$a_{5}$	$a_{5}$	0	$q_{5}$
6	0	0	−π2	$q_{6}$
7	0	0	π2	$q_{7}$
8	0	0	0	$q_{8}$
**End-Effector**				
9	$-l_{x}$	$l_{z}$	0	0
10	$-l_{y}$	0	0	0

**Table 2 sensors-22-08318-t002:** Similarity scores for the considered cases. *sim2* in both cases contains moments of the reconstructed signal from the EVAPose with the highest similarity values $\max\left(sim1\right)$.

	System Type	Type of Movement	30 FPS		Adjusted FPS	
			*sim1*	*sim2*	*sim1*	*sim2*
An ideal example, a laboratory environment	EVAPose	Up/down (vertical movement) (46s)—10 gestures	86.85	98.30	91.25 ^⊗^	98.30
		Left/right (horizontal movement) (46s)—10 gestures (single gesture measured in seconds)	82.3	92.45	89.65 ^⊗^	96.88
	OpenPose [42]	Up/down (vertical movement)	-	97.2	97.50 ^∅^	97.19
		Left/right (horizontal movement)	-	79.31	86.75 ^∅^	90.31
Real case from Gospoda ([47])	EVAPose	Example of up/down gesture (vertical movement) (19 gestures)	82.45	93.56	90.01 ^⊗^	93.56
		Example of a left/right gesture (horizontal movement) (12 gestures)	79.9	89.72	85.14 ^⊗^	89.72
	OpenPose [42]	Example of up/down gesture (vertical movement) (19 gestures)	-	95.01	95.33 ^∅^	96.12
		Example of a left/right gesture (12 gestures)	-	85.91	89.87 ^∅^	90.33

⊗ 19 FPS; ∅ 11 FPS.

## Data Availability

Not applicable.

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
