# Peer review of "Capturing Conversational Gestures for Embodied Conversational Agents Using an Optimized Kaneda–Lucas–Tomasi Tracker and Denavit–Hartenberg-Based Kinematic Model"

_sensors, 2022, doi:10.3390/s22218318_

Round 1

Reviewer 1 Report

The paper considers visual articulation of information presented by Embodied Conversational Agents that mimic human-like conversational behavior. The characteristic elements of the solution are signaled in the title. Additionally the Authors propose a measure based on cosine similarity to objectively evaluate the naturalness of synthesized hand movements, instead of subjective evaluation through human observation. They also present the results of the system evaluated objectively on an developed by them embodied conversational agent EVA.

The paper is interesting and convincing. To further improve it I suggest the Authors including a clear statement of the contribution. Moreover, please correct the caption of figure 4.  Optimal is the best or most favorable for a given situation, so most optimal is incorrect.

Author Response

Thank you for your comments on our article. We appreciate the time and effort you spent on  helping us improve this work. We hope that the following answers address all of your concerns sufficiently.

1: Regarding your comment '' To further improve it I suggest the Authors including a clear statement of the contribution.'':

A clear statement of the contribution was added to our article, find this in Introduction (line 99, in the last version of paper).

Statement of contribution:

With the goal of as natural as possible reconstruction of conversational gestures we present our choice of suitable methods, signaled in the title of this paper and our successful connection of stated methods in an efficient conversational gestures reconstruction system. We propose a measure based on cosine similarity for objectively evaluating the naturalness of by our method synthesized hand movements, instead of subjective evaluation through human observation, which is what, to the extent of our knowledge, was being done to evaluate gestures until now. In addition we present the results of our system evaluated objectively on an embodied conversational agent called EVA that we developed in our laboratory.

2: Regarding your comment '' Moreover, please correct the caption of figure 4.  Optimal is the best or most favorable for a given situation, so most optimal is incorrect.'':

Find most optimal changed to optimal in the caption of Figure 4 in the latest version of the article.  

Reviewer 2 Report

The article is very 'wordy' and could be significantly condensed. The solution to the described problem are mainly solved by reading the referred papers.

The references could be significantly reduced because many point to the same solution/problem.

Author Response

Thank you for your comments on our article. We appreciate the time and effort you spent on  helping us improve this work.

We shortened the article (from 21 to 20 pages) and removed some references (11 less). The amount of text was reduced to the extent that the message was still preserved.

Reviewer 3 Report

The topic presented in interesting and relevant.

The concept cannot be considered completely original, but the paper contributes with work towards achieving the objectives stated from the conceptual description.

The references listed concerning ECAs could include more recent works.

It is hard to understand what was the real contribution of the authors apart from chaining together the different methods and tools listed. Is there any particular part that was originally created during this work?

It is an interesting paper from engineering perspective, but it would benefit if descriptions could be made more using standard approaches and description languages like SysML, UML to describe the overall architecture of system, which is not present in the explanation.

Formating of some figures must be improved.

Author Response

1: Regarding your comment '' It is hard to understand what was the real contribution of the authors apart from chaining together the different methods and tools listed. Is there any particular part that was originally created during this work?'':

I am sorry if we weren't clear about our contribution. We have now included a clear statement of contribution in our paper, find this in (line 99, in the last version of paper).

As you stated, our proposed system is indeed a (otherwise original) combination of already known methods, enriched with a new, original method for completely objective assessment of the naturalness of gestures. The authors estimate that the work can represent an important contribution to our research area.

Statement of contribution:

With the goal of as natural as possible reconstruction of conversational gestures we present our choice of suitable methods, signaled in the title of this paper and our successful connection of stated methods in an efficient conversational gestures reconstruction system. We propose a measure based on cosine similarity for objectively evaluating the naturalness of by our method synthesized hand movements, instead of subjective evaluation through human observation, which is what, to the extent of our knowledge, was being done to evaluate gestures until now. In addition we present the results of our system evaluated objectively on an embodied conversational agent called EVA that we developed in our laboratory.

2: Regarding your comment '' It is an interesting paper from engineering perspective, but it would benefit if descriptions could be made more using standard approaches and description languages like SysML, UML to describe the overall architecture of system, which is not present in the explanation.''

We prepared the Figure 1 in accordance with the UML markup language but in our opinion the new version seems to be less transparent than the first one.

3: Regarding your comment ''Formating of some figures must be improved.'':

We have changed the formatting of some figures, on which in our opinion the labels on the axes were not perfectly visible. The resolution of the figures has also been fixed. If you think any other figures need changes, please let us know.